
# Mexican agricultural soil dust as a source of ice nucleating particles

Diana L. Pereira[1,2], Irma Gavilán[3], Consuelo Letechipía[4], Graciela B. Raga[1], Teresa Pi Puig[5], Violeta Mugica-Álvarez[6], Harry Alvarez-Ospina[7], Irma Rosas[1], Leticia Martinez[1], Eva Salinas[1], Erika T. Quintana[8], Daniel Rosas[9], and Luis A. Ladino[1]

[1]Instituto de Ciencias de la Atmósfera y Cambio Climático, Universidad Nacional Autónoma de México, Mexico City, Mexico
[2]Posgrado de Ciencias Químicas, Universidad Nacional Autónoma de México, Mexico City, Mexico
[3]Facultad de Química, Universidad Nacional Autónoma de México, Mexico City, Mexico
[4]Unidad Académica de Estudio Nucleares, Universidad Autónoma de Zacatecas, Zacatecas, Mexico
[5]Instituto de Geología & LANGEM, Universidad Nacional Autónoma de México, Mexico City, Mexico
[6]Universidad Autónoma Metropolitana- Azcapotzalco, Mexico City, Mexico
[7]Facultad de Ciencias, Universidad Nacional Autónoma de México, Mexico City, Mexico
[8]Escuela Nacional de Ciencias Biológicas, Instituto Politécnico Nacional, Mexico City, Mexico
[9]Facultad de Química, Universidad Autónoma de Yucatán, Mérida, Mexico

*Correspondence to:* Luis A. Ladino (luis.ladino@atmosfera.unam.mx)

**Abstract.** Agricultural soil erosion, both mechanical and eolic, may impact cloud processes as some aerosol particles are able to facilitate ice crystals formation. Given the large agricultural sector in Mexico, this study investigates the ice nucleating abilities of agricultural dust collected at different sites and generated in the laboratory. The immersion freezing mechanism of ice nucleation was simulated in the laboratory via the Universidad Nacional Autónoma de México (UNAM)- Micro Orifice Uniform Deposit Impactor (MOUDI)- Droplet freezing technique (DFT) (UNAM-MOUDI-DFT). The results show that agricultural dust from the Mexican territory promote ice formation in a temperature range from -11.8°C to -34.5°C, with ice nucleating particle (INP) concentrations between 0.11 $L^{-1}$ and 41.8 $L^{-1}$. Furthermore, aerosol samples generated in the laboratory are more efficient than those collected in the field, with $T_{50}$ values (i.e., the temperature at which 50% of the droplets freeze) higher by more than 2.9°C. The mineralogical analysis indicated a high concentration of feldspars i.e., K-feldspar and plagioclase (> 40%) in most of the aerosol and soil samples, with K-feldspar significantly correlated with the $T_{50}$ of particles with sizes between 1.8 µm and 3.2 µm. Similarly, the organic carbon (OC) was correlated with the efficiency of aerosol samples from 3.2 µm to 5.6 µm and 1.0 µm to 1.8 µm. Finally, a decrease in the efficiency as INPs, after heating the samples at 300°C for 2 h, evidenced that the organic matter from agricultural soils can influence the role of INPs in mixed-phase clouds.

## 1 Introduction

Agricultural activities may influence our environment and human health through the emission of aerosol particles (Telloli et al., 2014; Chen et al., 2017; Tomlin et al., 2020). It has been estimated that agricultural dust particles may represent between 25% (Ginoux et al., 2012) and 50% (Mahowald et al., 2004) of the global airborne dust. Dust particles impact visibility and water quality (Presley and Tatarko, 2009). Moreover, dust



particles influence global climate by affecting the Earth's radiative balance and cloud microphysical properties (Haywood and Boucher, 2000; Lagzi et al., 2013).

Aerosol particles influence cloud properties due to their ability to act as ice nucleating particle (INP) or cloud condensation nuclei (CCN) (Cotton and Yuter, 2009). INPs promote ice formation in clouds, influencing precipitation development and the hydrological cycle (DeMott et al., 2010). The predominance of precipitation formation via the ice phase over the continents makes the presence of ice particles fundamental (Mülmenstädt et al., 2015), especially in mixed-phase clouds, where immersion freezing has been reported as the main ice 50 formation pathway (Murray et al., 2012; Hande and Hoose, 2017). Immersion freezing allows ice activation from INPs embedded in liquid droplets when the temperature decreases below 0°C (Murray et al., 2012).

Given that soils are complex mixtures of mineral and organic components, living organisms, air, and water (Kalev and Toor, 2018), a wide variety of aerosol particles with potential INP abilities can be emitted during 55 agricultural activities, such as tilling (Steinke et al., 2016; Chen et al., 2017). Numerous studies have analyzed the ice nucleating abilities of the mineral components of dust particles (e.g., Eastwood et al., 2008; Zimmermann et al., 2008; Welti et al., 2009; Yakobi-Hancock et al., 2013; Broadley et al., 2012; Hiranuma et al., 2015; Boose et al., 2016), where its efficiency as INPs was found to be usually associated to the K-feldspar content (e.g., Yakobi-Hancock et al., 2013; Atkinson et al., 2013; Kiselev et al., 2016). On the other hand, Lee et al. (2006) 60 and Tomlin et al. (2020) observed that the concentration of microorganisms is enhanced in agricultural soil samples. Thus, a variety of microorganisms (e.g., bacteria, fungi) or their residues can be likely attached on mineral dust surfaces improving their ice nucleating abilities or they can also act as INPs by themselves (Conen et al., 2011; Després et al., 2012).

Biological particles have been reported as one of the most efficient INPs (Schnell and Vali, 1972; Hoose and Möhler, 2012; Hader et al., 2014; Kanji et al., 2017), with activation temperatures as high as -3°C (Christner et al., 2008; Huang et al., 2020). For example, Conen et al. (2011) observed that biological particles influence ice formation in soil samples at temperatures >-12°C. Garcia et al. (2012) reported ice nucleation active (INA) bacteria in soil samples, responsible for the nucleation events observed at -12°C. Similarly, soils can serve as 70 an important source of organic particles (Montgomery et al., 2000; Kelleher and Simpson, 2006; Hill et al., 2016), which may be able to catalyze ice formation (Hoose and Möhler, 2012; Knopf et al., 2018; Chen et al., 2018; Chen et al., 2021). Knopf et al. (2018) have shown that Humic Acids (HA) and Humic-Like Substances (HULIS) influence ice nucleation in the immersion freezing and deposition nucleation modes. Chen et al. (2018) reported that droplets containing HULIS with concentrations from 15.8 to 96.7 mg L$^{-1}$ freeze at temperatures 75 between -9°C and -22°C. The ice nucleating abilities of HULIS may be a consequence of particles aggregation, which provides suitable surfaces for ice activation (Chen et al., 2021).

The ice nucleating abilities of agricultural soils from Argentina, England, Germany, Hungary, Mongolia, USA, and Russia have been evaluated in the last decades (Conen et al., 2011; Garcia et al., 2012; Tobo et al., 2014;





Steinke et al., 2016; Suski et al., 2018). Those studies suggest that the organic compounds in agricultural soils are more efficient at facilitating ice formation than the mineral components; however, Conen et al. (2011) highlighted the fact that the influence of organic components in ice formation is not considered in climate models. Also, it is important to note that although agriculture is an ubiquitous activity in tropical countries such as Mexico, the ice nucleating abilities of agricultural dust particles from the Mexican territory have not been

reported up to date.

Therefore, in the present study, the ice nucleating abilities of Mexican agricultural soils are evaluated for the first time using the Universidad Nacional Autónoma de México (UNAM)- MicroOrifice Uniform Deposit Impactor- Droplet Freezing Technique (UNAM-MOUDI-DFT), focusing on the influence of the mineralogical

composition versus the organic content and the size of the INPs.

## 2 Methods

### 2.1 Sampling sites

Airborne and soil samples were collected in 4 different Mexican states characterized by important agricultural

activities: Mexico City (CDMX), Morelos (MOR), Zacatecas (ZAC), and Yucatán (YUC), (Fig. 1). Soil samples and aerosol particles were collected at the ZAC site during a short-term field campaign between February 24 and 27, 2020. Soil samples from the CDMX, MOR, and YUC sites were collected in agricultural areas and provided through collaborations. The sampling points and the number of samples collected are summarized in Table 1, where samples are labeled based on the previous crop at each site.


Inside the mentioned states (CDMX, MOR, ZAC, YUC), specific municipalities (Table 1) containing different crops were analyzed. Milpa Alta is located at the southeast of CDMX and it is well known as the first nopal producer of the state (Alcaldía Milpa Alta, 2019). Totolapan (MOR) is a traditional maize (corn) producer in ~ 50% of its area (Ayuntamiento de Totolapan, 2018). Morelos (ZAC) is one of the main producers of chili, and

its agricultural area represents 25% of the territory, with some mining activities also reported (Covarrubias and Peña Cabriales, 2017). Hunucmá is located in YUC, where 22% of the territory is destined to agricultural activities (INEGI, 2015).

### 2.2 Field samples

Soil samples from the top 10 cm were collected in CDMX, MOR, ZAC, and YUC as shown in Table 1. In ZAC, soil samples were collected during soil tillage in addition to aerosol samples collected at the ground level. Aerosol particles were collected on siliconized glass substrates (HR3-215; Hampton Research) using a cascade impactor MOUDI (100R; MSP Corporation), which collects and classifies the particles according to their aerodynamic diameter (dp) onto eight stages (cut sizes of 0.18, 0.32, 0.56, 1.0, 1.8, 3.2, 5.6, and 10.0 µm), at a

flow of 30 L min$^{-1}$. Soil and aerosol samples were sealed and transported to the laboratory. Aerosol samples were stored at 3ºC and the soil samples at room temperature.





### 2.3 Laboratory generated samples

All soil samples were air dried, crushed, and sieved to a pore particle size of 425 μm. Aerosol particles were
then generated using a dry system (Fig. 2) based on the Ladino and Abbatt (2013) design. Briefly, the
experimental setup contains (a) an aerosol disperser consisting of a stirring plate and a metallic flask, on which
the particles are generated by turbulence, (b) a mixing flask to homogenize the samples, and (c) an aerosol
collector. A MOUDI 100R was used to collect aerosol particles for the analysis of INPs and the analysis of the
mineralogical composition, while a MiniVol TAS (Airmetrics) was used to collect aerosol particles for the
organic carbon (OC) analysis. In addition, an Optical Particle Counter (OPC) LasAir III (310 B; Particle
Measuring Systems) was used to obtain the particle size distribution (PSD) of the aerosol samples. The OPC
was operated at a flow rate of 28.3 L min$^{-1}$, and the aerosol concentrations were for particles ranging between
0.3 μm and 10 μm.

With the MOUDI, siliconized glass substrates (HR3-215; Hampton Research) and 47 mm aluminum foil filters
(0100-47-AF, TSI) were used for the INPs and mineral analysis, respectively. Aerosol particles collected over
aluminum filters on the eight stages of the MOUDI (0.18 to 10 μm) were grouped in a single sample for the
mineral analysis. Particulate matter with diameter less than 10 μm (PM$_{10}$) was collected over 47 mm quartz
filters (2500QAO-UP, Pall Life Science) using the MiniVol at a flow of 5 L min$^{-1}$ for the OC analysis. The
quartz filters were previously conditioned at 500°C for 4 h to remove trace pollutants, especially volatile organic
compounds.

### 2.4 Analysis of INPs

The ice nucleating abilities of the agricultural dust particles collected in the field and generated in the laboratory
were analyzed through the immersion freezing mode using the UNAM-MOUDI-DFT (Córdoba et al., 2021).
The equipment consists of: (a) a cold stage, (b) a humid/dry air system, (c) an optical microscope Zeiss Axio
Scope A1 (Axiolab Zeiss) with a recording system, and (d) a data acquisition system (Córdoba et al., 2021).
Briefly, the glass substrates containing the aerosol particles impacted on them were introduced and fixed in the
cold stage, at which the temperature was controlled by a thermostat (LAUDA PRO-RP 1090) filled with
polydimethylsiloxane. Afterward, humid air was directed towards the sample to allow liquid droplets formation
over the aerosol particles, until they reach a size of ca. 170 μm. Once most of the droplets have reached this
size, dry air (N$_2$) was used to evaporate droplets to avoid contact between them. Finally, the system was isolated,
and the temperature was decreased from 0°C to -40°C at a cooling rate of 10°C min$^{-1}$ to allow the freezing of
the droplets. The experiments were recorded, and the temperature was obtained with a resistance temperature
detector (RTD) connected to a Fieldlogger device (RS485, NOVUS) and placed at the center of the cold stage.
The recorded videos and the RTD temperatures allow the determination of the freezing temperature for each
droplet.

The INP concentrations (L$^{-1}$) were calculated using Eq. 1 (Mason et al., 2015a).




$$[INPs\,(T)] \; = \; -ln(\tfrac{N_u(T)}{N_o})N_o f_{nu,0.25-0.1mm} f_{ne}(\tfrac{A_{deposit}}{A_{DFT}V}) f_{nu,1\,mm} \tag{1}$$

where $N_u(T)$ is the number of unfrozen droplets (dimensionless) at a temperature $T$ (°C), $N_o$ is the total number of droplets (dimensionless), $f_{ne}$ is a correction factor that accounts for the uncertainty associated with the number of nucleation events (dimensionless), $A_{deposit}$ is the total area of the aerosol deposit on the glass substrates (mm$^2$), $A_{DFT}$ is the area of the glass substrates analyzed by the DFT (mm$^2$), $V$ is the volume of air sampled with the MOUDI (L), and $f_{nu}$ are correction factors to account for the aerosol deposit inhomogeneity (dimensionless).

**2.5 Analytical Techniques**

X-ray diffraction (XRD) has been widely used for the characterization of crystalline materials (Kohli and Mittal, 2019). Therefore, the mineralogical composition of the aerosol and soil samples were determined using a X-ray diffractometer Empyrean (Malvern Panalytical, with CuKα radiation) operated with a PIXcel 3D detector. The mineral phases were identified and quantified by the Rietveld method (Rietveld, 1969) using the HIGHScore v4.5 software and ICDD (International Center for Diffraction Data) and ICSD (Inorganic Crystal Structure Database) databases.

The OC content was derived using a thermal-optical technique (Sunset Lab) based on Birch and Cary (1996) procedure. Briefly, the quartz filters were introduced in an oven, where the samples were volatilized and oxidized to $CO_2$. Finally, the $CO_2$ was reduced and quantified by a flame ionization detector. To verify the influence of the organic matter in the ice nucleating abilities of the agricultural dust, the soil samples were heated at 300ºC for 2 h to remove the organic components, following Tobo et al. (2014).

**2.6 Microbiological Analysis**

To determine culturable microorganisms present on the soil samples collected in ZAC, 500 mg of each sample were added in 10 mL of sterile solution at 0.85%. After 1:100 dilution and vortex agitation, 0.1 mL of solutions were cultured on three growing media such as Trypticase Soy Agar (TSA) and MacConkey Agar (MCA) between 24 and 48 h, and Malt Extract Agar (MEA) for 3 days. The TSA and MCA growing media were used to cultivate bacteria and the MEA for fungal propagules. Then, the colony forming units (CFU) were obtained for 1.0 g of soil.

**3 Results and Discussion**

**3.1 Ice nucleating abilities**

Figure 3 summarizes the ice nucleating abilities of the different agricultural dust particles with sizes between 0.56 µm and 5.6 µm, corresponding to the MOUDI stages 3 to 6. The present results focus on particles >0.56 µm as it has been shown that particles >0.5 µm have a higher potential to act as INPs (e.g., DeMott et al., 2010). Two sets of samples are presented in Fig. 3: aerosol samples collected directly in the field (F) and those





generated in the laboratory (L) from top-soil collected in the fields. The L samples (solid lines) acted as INPs between -11.0ºC and -26.0ºC, while the F samples (dotted lines) show a much wider temperature range (i.e.,

between -11.8ºC and -34.5ºC). In terms of the $T_{50}$ (i.e., the temperature at which the 50% of the droplets freeze), the highest and lowest mean values for the L samples were -19.7ºC (for dp=1.8-3.2 µm) and -20.7ºC (for dp=3.2-5.6 µm). Similarly, for the F samples the highest mean $T_{50}$ was -23.4ºC (for dp=1.8 to 3.2 µm) and the lowest -26.1 ºC (for dp=0.56 to 1.0 µm). Therefore, the L samples were found to nucleate ice at higher temperatures compared to the F samples.


The warmest freezing temperatures shown by L samples suggest that the aerosol particle generation during soil tillage is not fully simulated by the process used in the laboratory. The discrepancies in the INP abilities can be attributed to different environmental conditions as the F samples are exposed to a variety of physicochemical processes while in the atmosphere (Boose et al., 2016; Cziczo et al., 2013), which is unlikely the case in the L

samples. The differences between laboratory and field environments are also reflected in different PSD observed during the aerosolization process (Fig. S1). As this figure shows, mean particle concentrations between $5.0 \times 10^{-4}$ and 0.4 particles cm$^{-3}$ characterized L samples, while lower values were observed for the F samples. Furthermore, the highest particle concentration for the L samples was found for particles between 1.0 µm and 5.0 µm (Fig. S1a), while the F samples are enriched in smaller particles, i.e., 0.3 µm (Fig. S2b).

Therefore, the larger particles present in the L samples likely promoted ice nucleation at warmer temperatures.

It was also found that the ice nucleating abilities of the different soils seem to be influenced by the specific crop grown previously and the type of soil, with particles from the nopal, corn 1, and corn 2 crops showing the warmest freezing temperatures and beans and wheat showing the coldest freezing temperatures (Fig. 3). As

Table 1 shows, the ZAC samples were collected in calcisols; however, the concentrations of the mineral phases identified on bean, chili, wheat, and onion samples differ, suggesting that additional parameters to the soil type may influence the samples properties and their abilities to act as INPs. Kalev and Toor (2018) found that soil composition determines their properties. This fact may influence the ice nucleating abilities of the soils, as shown in Fig. 3. Further details of the mineral composition and the organic content of each sample are discussed

below.

Tegen and Fung (1995) and Tegen et al. (2004) also proposed that the PSD of the aerosol particles can vary according to soil type. The size of the aerosol particles is well known to influence their behavior as INPs (Diehl and Wurzler, 2004; DeMott et al., 2010; Mason et al., 2015b; Córdoba et al., 2021). This fact is evidenced in

the different PSD distributions observed for each sample in Fig. S1. Figure S2 shows a clear trend for the F samples, where the larger the particle size, the higher the $T_{50}$. However, this behavior was not observed for the L samples, corroborating differences in the PSD between L and F samples.

Although the laboratory generated aerosol particles do not fully reproduce the characteristics of the ambient

agricultural particles, the ice nucleation experiments of the L samples highlight the importance of agricultural



soils in ice formation in Mexico. The ice nucleation temperatures observed in the present study are on the same order as those reported for agricultural dust in Wyoming (USA), from -18ºC to -36ºC for dp=0.6 μm (Tobo et al., 2014), and Argentina, China, and Germany from -11ºC to -26ºC for dp <5 μm (Steinke et al., 2016). This suggests that they are able to influence ice formation in clouds regardless of the origin or location of the agricultural soils. The comparison of the present results with literature data is further discussed in Sect. 3.4.

### 3.2 The influence of organic matter

Figure 4 shows that the OC concentration represents a small fraction of the agricultural dust samples (dp <10 μm), with values between 5% and 17%, while the mineral components predominate (i.e., from 33% to 95%). Conen et al. (2011) found that the OC fraction for non-agricultural soils with dp <15 μm collected in Mongolia, Germany, Hungary, and Russia varied between 0.7 and 12%. O'Sullivan et al. (2014) also reported small values of OC, from 2 to 13% for agricultural dust (bulk) in England. In contrast, Tobo et al. (2014) measured higher concentrations of organic compounds (37%) for agricultural dust dp <0.6 μm in the US. The aforementioned studies reported that the organic components of soil dust (with different particle sizes and concentrations) can enhance the ice nucleating abilities of dust particles. This behavior is also observed for the agricultural dust analyzed here, as summarized in Fig. 5.

Figure 5 shows that for the four particle sizes ranges analyzed here (i.e., 0.56-1.0 μm, 1.0-1.8 μm, 1.8-3.2 μm, and 3.5-5.6 μm), there is a significant reduction in the freezing temperature, referred as $\Delta T_{50}$, after the organic matter was degraded through a heating treatment. The $T_{50}$ of the heated samples got reduced between 0.7ºC and 14.0ºC, with the largest mean $\Delta T_{50}$ reported by particles in the size range between 1.8 and 3.2 μm, as shown in Fig. S3. The highest reduction in the ice nucleating abilities as a consequence of the heat treatment, was observed on the corn 2 sample ($\Delta T_{50}$=-14ºC, Table S1) with 17% of OC, while the other samples have OC values <9% (Fig. 4). The highest reduction was observed for particles with sizes ranging between 1.8 μm and 3.2 μm, which were also reported as the most efficient INPs (Fig. 3). Even though the OC fraction is small compared to the mineral components, the reduction in freezing temperatures of the soil samples analyzed here suggests that the organic components present in the samples increase their ice nucleating abilities. This behavior has been widely observed after the removal of the organic matter (O'Sullivan et al., 2014; Tobo et al., 2014; Suski et al., 2018) or the destruction of proteinaceous compounds (Conen et al., 2011; Garcia et al., 2012; Steinke et al., 2016) from agricultural soils. Organic matter on soils may come from microorganisms and/or their residues, humic (e.g., humic acids) and non humic (e.g., proteins, polysaccharides) substances, and refractory compounds (Hill et al., 2016).

Different types of organic matter were not analyzed here; however, the presence of microorganisms as a possible source of organics in the ZAC soil samples was confirmed, as shown in Fig. S4. Microorganism concentrations were observed to be as high as $5.6x10^6$, $1.98x10^5$, and $2.4x10^4$ CFU $g^{-1}$ for mesophilic bacteria, fungal propagules, and gram-negative bacteria, respectively. The concentrations of airborne microorganisms have been reported during harvesting activities in the US (Lighthart, 1983; Lee et al., 2006). Lighthart et al. (1983)



observed that airborne concentrations of fungi and bacteria can reach values as high as $10^9$ CFU m$^{-3}$. During corn harvesting, concentrations between $3.5 \times 10^5$ and $1.4 \times 10^6$ CFU m$^{-3}$ were observed for culturable bacteria and between $1.6 \times 10^6$ and $7.4 \times 10^6$ CFU m$^{-3}$ for culturable fungal spores (Lee et al., 2006). Although direct intercomparison cannot be performed between these studies, the observed airborne concentrations of microorganisms suggest that an important fraction of microorganisms containing in soils can be aerosolized during soil manipulation. Furthermore, it is important to highlight the presence of gram-negative bacteria found in the ZAC agricultural soils, as those are known to be active as INPs (Šantl-Temkiv et al., 2015)

As soils contain more than organic compounds, the effects of heat treatments on minerals have been questioned. Tobo et al. (2014) show that the ice nucleating abilities of mineral samples are not affected by heat treatments during two hours at 300ºC, and Perkins et al. (2020) observed a similar behavior for the mineral dust proxy Arizona Test Dust (ATD) at 500ºC. Although those studies suggested that minerals are not strongly affected by dry heat treatments, recent observations by Daily et al. (2021) show that shifts in minerals efficiency as INPs cannot be neglected, as heat treatments at 250ºC might slightly deactivate K-feldspar compounds. This is not the case of the corn 2 sample (Table S1), which lacks feldspars and highlights the importance of organic compounds at temperatures >-20ºC.

Note that the decrease observed in freezing temperatures varies as a function of particle size (Figs. 5 and S3), suggesting a relationship between the organic content and the size of the particles. These observations agree with Chen and Chiu (2003) and Lin et al. (2010), who reported that the composition of the organic matter contained in soils from Taiwan and the HULIS fraction from China soils varies with particle size. In addition, the positive significant correlations found between the $T_{50}$ and the OC concentration for particles ranging between 1.0 and 1.8 µm (r=0.79, p-value<0.05) and between 3.5 and 5.6 µm (r=0.86, p-value<0.05) shown in Fig. S5, support the importance of particle size and chemical composition in the ice nucleating abilities of agricultural dust particles.

**3.3 Mineralogical analysis**

The presence of plagioclase, K-feldspar, quartz, kaolinite, smectite, mica-illite, and minor constituents were identified through XRD analysis. Figure 6 shows a high content of feldspars (i.e., K-feldspar, plagioclase) in both soil and aerosol samples, except in the corn 2 sample collected in Hunucmá, where the presence of feldspars was not identified. In particular, the plagioclase fraction (i.e., Na/Ca feldspar) together with the K-feldspar reaches more than 50% of the total concentration for the corn, nopal, and bean samples. The absence of feldspars compounds in Hunucmá sample can be a consequence of the soil type, as leptosols are enriched in calcareous materials (SEMARNAT, 2002).

K-feldspar is well known for its efficiency as INP (Atkinson et al., 2013; Peckhaus et al., 2016; Kiselev et al., 2016), mostly associated with the presence of active sites (i.e., cracks, defect, cavities) contained on its surface (Kiselev et al., 2016). Therefore, the high concentrations of feldspars in the Mexican agricultural soils can also





influence their ice nucleating abilities, as was observed in the high and statistically significant correlation between the K-feldspar concentration and the $T_{50}$ of aerosol particles with sizes ranging between 1.8 and 3.2 μm (r=0.85, p-vale<0.05) as shown in Fig. S5. Boose et al. (2016) observed a similar behavior for dessert

samples and the presence of feldspars as they found good correlations between compounds containing feldspars and INPs efficiency.

Different mineral phases between the samples can be the source of the different ice nucleating abilities, as shown in Fig. 3. The relationship between aerosol particle composition and their ice nucleating abilities has

been previously reported (Baustian et al., 2012; O'Sullivan et al., 2014; Paramonov et al., 2018; Steinke et al., 2020; Hiranuma et al., 2021). Furthermore, the differences in the mineralogical composition between the soil and aerosol samples indicate that not all the soil particles are aerosolized. These differences can also be a consequence of the particle sizes analyzed, as aerosol samples varied between 0.18 and 10 μm, while the soils particles were smaller than 425 μm.


The presence of feldspars, quartz, and clays (e.g., illite, kaolinite) derived from XRD analysis have been commonly identified in mineral dust transported from Africa (Broadley et al., 2012) and other desserts around the world (e.g., Australia, Atacama, Boose et al., 2016). Similar mineral phases such as kaolinite, illite, quartz, and minor concentrations of feldspars, were also observed in agricultural soils from England (O'Sullivan et al.,

2014). Therefore, the present results and those previously reported evidence that although there are differences in the type and origin of the soils, they can have similar mineral components that determine the ice nucleation behavior of their aerosol particles.

### 3.4 INP concentrations and atmospheric implications

The INP concentration emitted during soil tillage of the agricultural soils in ZAC was found to vary between 0.11 $L^{-1}$ and 41.8 $L^{-1}$ from -11.8ºC to -34.5ºC as shown in Fig. 7. The INP concentrations measured in ZAC are in agreement with those reported for Colorado (USA) (Garcia et al., 2012), England (O'Sullivan et al., 2014), Kansas (USA) (Mason et al., 2016; Suski et al., 2018), and Wyoming (Tobo et al., 2014; Steinke et al., 2020). Garcia et al. (2012) reported INP concentrations between $7.8 \times 10^{-3}$ $L^{-1}$ (-7.0ºC) and 5.5 $L^{-1}$ (-20.2ºC) derived

with a Droplet Freezing Assay (DFA). Using the same method, O'Sullivan et al. (2014) observed INP concentrations varying from $3.6 \times 10^{-6}$ $L^{-1}$ (-6.2ºC) to $4.4 \times 10^{2}$ $L^{-1}$ (-26.1ºC). Steinke et al. (2020) found INP concentrations between 0.15 $L^{-1}$ (-18.2ºC) and $2.5 \times 10^{3}$ $L^{-1}$ (-28.2ºC) using the Aerosol Interactions and Dynamics in the Atmosphere (AIDA) cloud chamber. Tobo et al. (2014) reported INP concentrations, analyzed in the Colorado State University Continuous Flow Diffusion Chamber (CSU-CFDC), between $10^{-2}$ $L^{-1}$ (-18ºC)

and $10^{4}$ $L^{-1}$ (-36ºC) for soils with a 25% of agricultural origin and 75% of desert origin. Suski et al. (2018) implemented an Ice Spectrometer and found INP concentrations from $3.4 \times 10^{-3}$ $L^{-1}$ to $8.5 \times 10^{2}$ $L^{-1}$ at temperatures between -5.5ºC and -25.5ºC. Using a similar setup to the present study, Mason et al. (2016) derived total INP concentrations at Colby (Kansas) of 0.5 $L^{-1}$, 1.0 $L^{-1}$, and 8.9 $L^{-1}$ at -15ºC, -20ºC, and -25ºC, respectively.



Although different devices have been used to determine the INP concentrations in agricultural dust samples, with differences in their detection limits and sensitivities, the INP concentrations reported here are within the same range, as shown in Fig. 7. The similarities between the present results and previous studies evidence the importance of the mineral and organic components of soils dust in ice formation, with organic components increasing the INP efficiency of soil particles, as observed in Sect. 3.2.


In agricultural areas, the organic components appear as fundamental for the inherent capacity of soils to facilitate ice formation, even at low concentrations (Conen et al., 2011; O'Sullivan et al., 2014). This fact suggests that more attention must be paid to determine specific organic components from soils and to understand their individual ice nucleating abilities. This efficiency can also be influenced by the size of the particles, as

shown in Figs. 3 and S1. Furthermore, the present results show that emissions from tropical latitudes are consistent with those reported for mid and high latitudes. Agricultural lands cover ca. 13% of the total surface of Mexico (Torres and Martínez, 2019), and represent the economic activity with the highest income within the primary sector (CEDRSSA, 2021). Given that the economy of several tropical countries is based on agricultural activities, it is very important to determine the role that their associated emissions play in the local and regional

hydrological cycle. Also, agricultural dust particles must be included in numerical studies aiming to predict future climate as the food demand will increase, and hence, the land used for food production.

**4 Conclusions**

This study reports for the first time the ice nucleating abilities of Mexican agricultural dust via immersion freezing. INP concentrations between 0.11 $L^{-1}$ and 41.8 $L^{-1}$ in the temperature range from -11.8ºC to -34.5ºC were observed. The measured concentrations are comparable to those reported for agricultural soils in the United States and England, confirming the contribution from tropical emissions and their role in mixed phase cloud formation. The comparison of ice nucleating abilities of aerosol samples aerosolized in the laboratory and

collected at the field indicate a higher efficiency as INPs in laboratory samples, with freezing temperature values varying between -11ºC and -26ºC and $T_{50}$ values higher by more than 2.9ºC, as a consequence of higher particle concentrations of larger particles.

The XRD analysis allowed the identification of the different mineral phases present in the aerosol and soil

samples, where high concentrations of plagioclase, K-feldspar, quartz, kaolinite, smectite, and mica-illite were detected. These minerals have been previously identified in dust samples. In particular, feldspars were found in higher concentrations (> 40%) for most of the samples. Additionally, the significant correlation between the $T_{50}$ and the K-feldspar for particles with sizes from 1.8 to 3.2 µm shows the influence of K-feldspar in the INPs efficiency on mineral particles. The concentrations of OC indicate that despite the low percentage observed in

most of the samples (<17%) in comparison to the mineral concentration, the organic components increase the efficiency as INPs to promote ice crystals formation. This is evidenced by the decrease in the efficiency of the



ice nucleating abilities after the removal of the organic matter, and the statistically significant correlations between the OC concentration and the $T_{50}$, for particle size from 3.2 to 5.6 µm and 1.0 to 1.8 µm.

The present results improve the current gap in knowledge of field measurements of aerosol particles at tropical latitudes, focusing on agricultural emissions and highlights the importance of both the chemical composition and the particle size in their efficiency as INPs. However, more analysis and especial attention to the organic compounds of agricultural dust are needed to improve the current understanding of soil components and the development of new parametrizations.


**Data availability.** Data are available upon the request to the corresponding author.

**Author contribution.** LAL, GBR, and IG designed the field campaigns and the experimental process. DLP, LAL, CL, EQ, and DR performed the field measurements. DLP, LAL, TP, VM, HA, IR, LM, and ES carried
out the experiments. DLP and LAL wrote the paper with contributions from all coauthors.

**Competing interests.** The authors declare that they have no conflict of interest.

**Acknowledgments**
The authors thank Fernanda Córdoba, Luis Gonzales, and Dara Salcedo for their helpful support. This work was funded by the Consejo Nacional de Ciencia y Tecnología (CONACYT), through the FC-2164 and CB-285023 grants. DLP also thank CONACYT for the Master's scholarship.

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





**Table 1:** Summary of the details of each aerosol and soil sample collected at the field in four different Mexican states: Mexico City (CDMX), Morelos (MOR), Zacatecas (ZAC), and Yucatán (YUC).

| Sample | Soil type | Sampling Place | Latitude (°N) | Longitude (°O) | N° soil samples | N° aerosol samples |
|--------|-----------|----------------|---------------|----------------|-----------------|--------------------|
| Nopal | Andosol | Milpa Alta, CDMX | 19.1991 | 99.0170 | 1 | 0 |
| Corn | Andosol | Totolapan, MOR | 19.0019 | 98.9161 | 1 | 0 |
| Bean | Calcisol | Morelos, ZAC | 22.8050 | 102.6750 | 1 | 1 |
| Chili | Calcisol | Morelos, ZAC | 22.8380 | 102.6853 | 1 | 1 |
| Wheat | Calcisol | Morelos, ZAC | 22.8508 | 102.6476 | 1 | 1 |
| Onion | Calcisol | Morelos, ZAC | 22.8164 | 102.6791 | 1 | 1 |
| Corn 2 | Leptosol | Hunucmá, YUC | 20.9999 | 89.8575 | 1 | 0 |

The soil type was derived from the SAGARPA (2017).


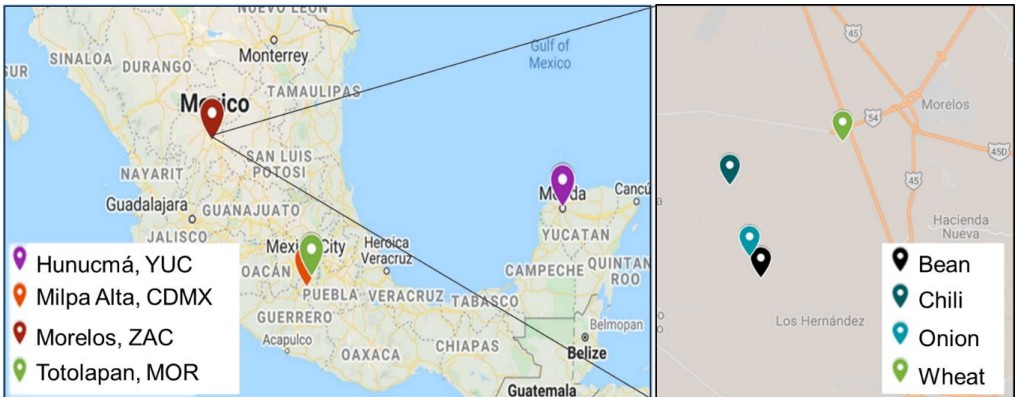

**Figure 1:** Map showing the sampling sites along four different states of Mexico: Mexico City (CDMX), Morelos (MOR), Zacatecas (ZAC), and Yucatán (YUC). The zoom in shows the ZAC sampling spots with their corresponding crop (© Google Maps, 2021).






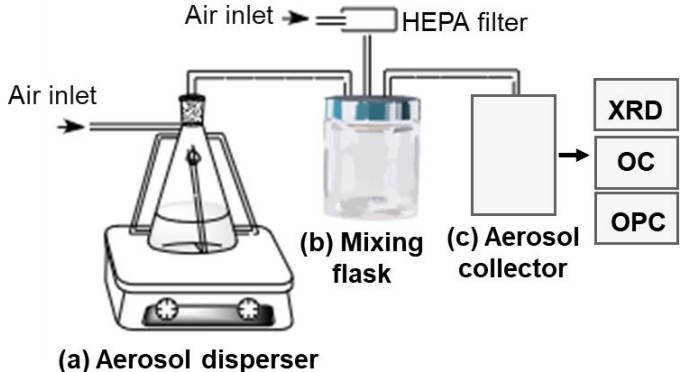

**Figure 2:** Dry aerosol generation system setup. (a) aerosol disperser, (b) mixing volume, and (c) aerosol collector (MOUDI 100R or MiniVol TAS). XRD, OC, and OPC refers to X-ray diffraction, organic carbon, and optical particle counter, respectively.

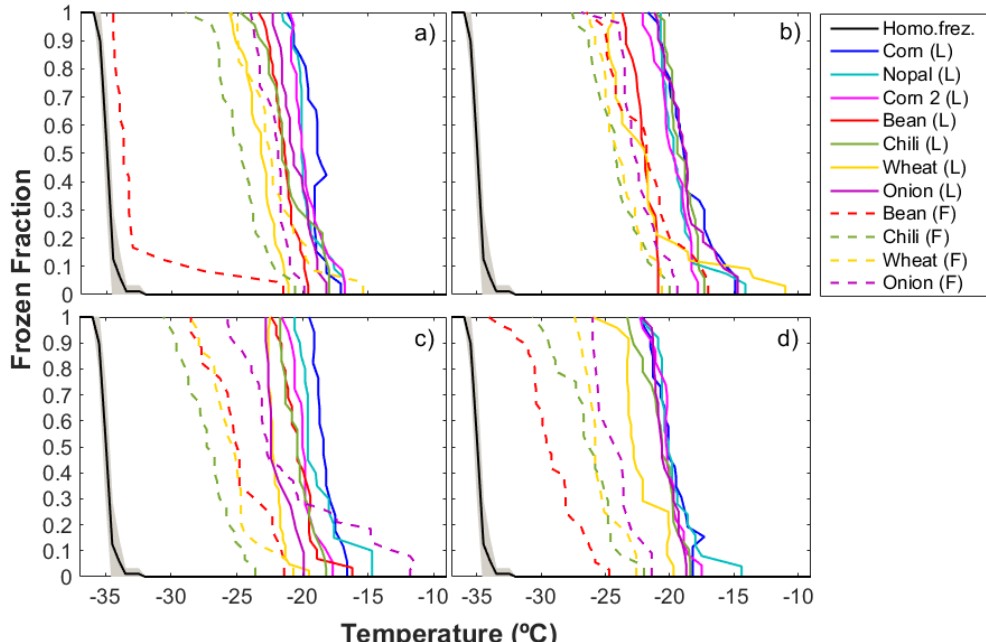


**Figure 3:** Ice nucleating abilities of the agricultural soil particles as a function of temperature and particle size a) 3.2-5.6 µm, b) 1.8-3.2 µm, c) 1.0-1.8 µm, and d) 0.56.1.0 µm. The black line (Homo. frez.) depicts the average homogeneous freezing curve. The dotted and continuous lines show the results of the samples collected at the field (F) and the samples generated in the laboratory (L), respectively.






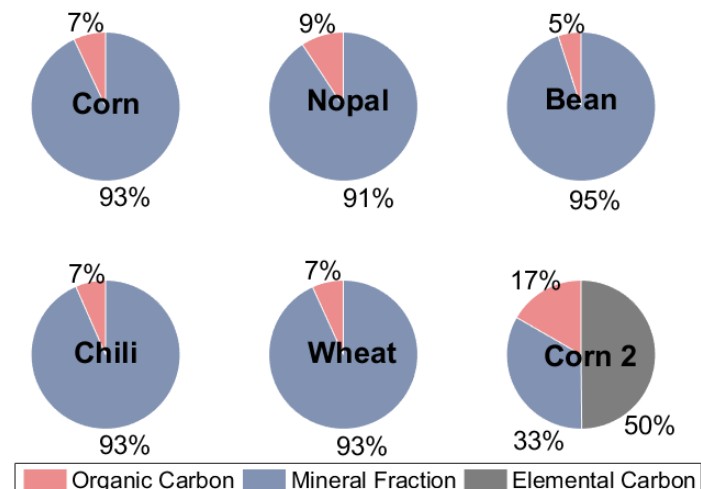

**Figure 4:** Organic and mineral contribution of the aerosol samples generated in the laboratory.

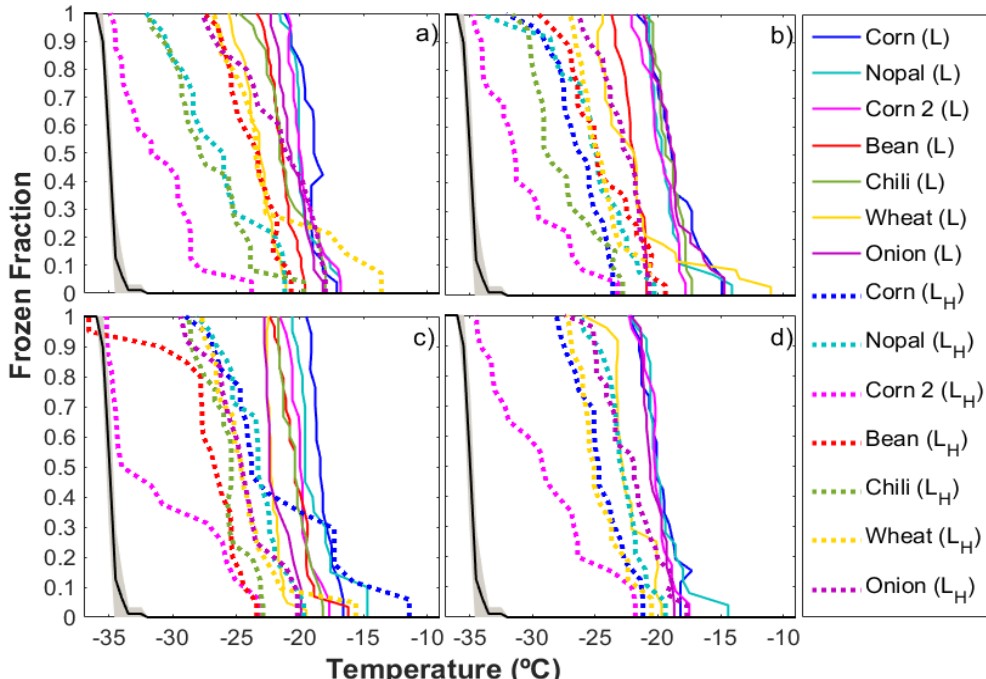

**Figure 5:** Ice nucleating abilities of agricultural dust particles generated in the laboratory (L) before and after

the heating treatment as a function of temperature and particle size a) 3.2-5.6 μm, b) 1.8-3.2 μm, c) 1.0-1.8 μm, and d) 0.56.1.0 μm. The black line (Homo. frez.) depicts the average homogeneous freezing curve. The dotted and continuous lines show the results of the heated samples (H) and non-heated samples, respectively.





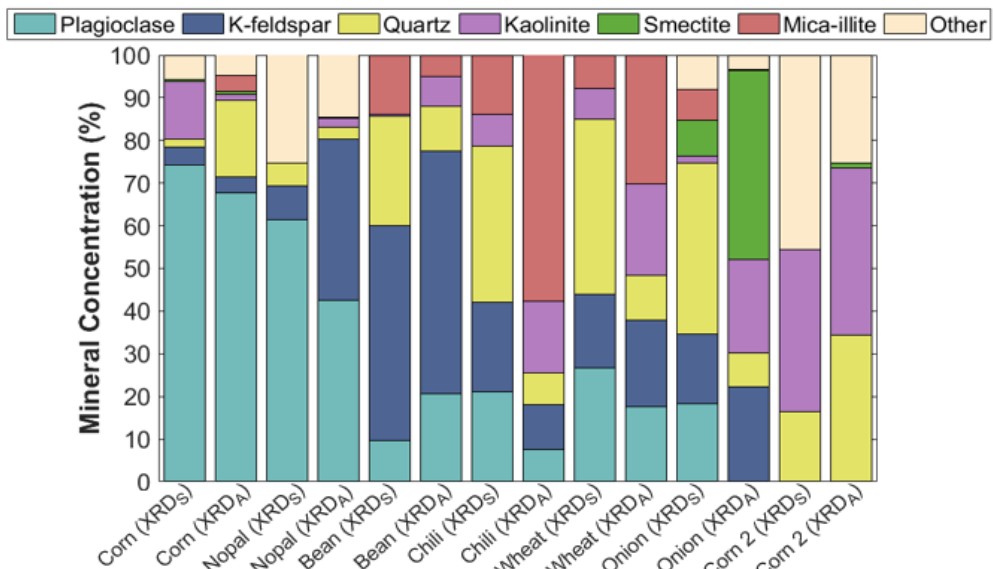

**Figure 6:** Summary of the mineralogical composition of the aerosol particles generated in the laboratory for dp=0.18 to 10 μm (XRD$_A$), and soil samples collected in the field for particle size <425 μm (XRD$_S$). The mineral phases were identified using X-ray diffraction (XRD).

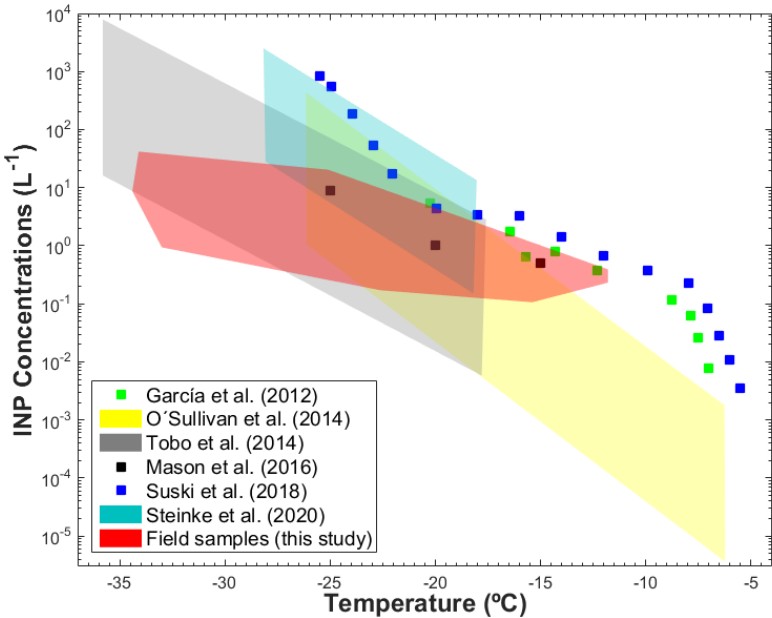

**Figure 7:** Inter-comparison of the INP concentrations as a function of temperature from the present study (red area) and literature data (blue, gray, and yellow area, and filled symbols).