# Peer review of "Mexican agricultural soil dust as a source of ice nucleating particles"

_Atmospheric Chemistry and Physics, 2021_

## Author Comment (AC1)

We would like to thank the Reviewers for their constructive suggestions, which helped us to improve the manuscript. Specific answers and manuscript modifications related to the Reviewer's comments are given below in red text. The line numbers correspond to the revised manuscript with track-changes.

**REFEREE 1**

**General Comments**

This work evaluates the ice nucleating ability of Mexican agricultural soil dust and discuss the possible factors influencing its ice nucleating ability. Since "the ice nucleating abilities of agricultural dust particles from the Mexican territory have not been reported up to date (Lines 84-85)", the dataset in this area will be potentially helpful to improve our knowledge of the possible impact of agricultural soil dust on ice nucleation in mixed-phase clouds. Unfortunately, however, there are some problems regarding the experimental approaches. For example, it is not clear when and how the soil and aerosol samples were obtained. The use of sized-resolved INP data are unique, but it is disappointing that only limited INP data in a narrow size range are reported in this manuscript. Although this work tries to evaluate the possible importance of organic and mineral components in the agricultural soil dusts using several approaches, the conclusions are obscure and not well summarized. In addition, it is difficult to compare the results presented in this manuscript with those from other previous studies which evaluated the ice nucleating ability of agricultural soil dusts in other locations, because this study uses $T_{50}$ values while most studies have used other parameters like the ice-nucleation active site density (INAS). For these reasons, it is almost impossible to evaluate the quality of the data and whether the key conclusions presented here are scientifically appropriate. I think that the quality of this manuscript should be significantly improved before considering publication.

A: We thank the reviewer for her/his careful evaluation of our manuscript. We are confident of the experimental procedures used and are aware of the technical limitations. We hope that the explanations provided below, together with the additional information and data added to this document, will clarify the study for the reviewer to allow assessment of the quality of the data. We believe that our conclusions are robust and supported by the provided information, highlighting the importance of agricultural emissions in mixed-phase cloud formation.

**Specific Comments**

1) It is not clear when the soil and aerosol samples were collected. As for the soil samples, please clarify when the samples were collected in Table 1. As for the aerosol samples, please clearly explain when and where each sampling was started and finished (this information should be summarized in a table).

A: Thank you for this comment, which we now clarify. Sampling dates were added to the revised Table 1. Additionally, the meteorological information of the Zacatecas campaign was added into the supplementary information of the revised manuscript as Table S1.

**Table 1:** Summary of the details of each aerosol and soil sample collected in four different Mexican states: Mexico City (CDMX), Morelos (MOR), Zacatecas (ZAC), and Yucatán (YUC). The samples were labeled based on the previous crop present in the sampling location.

| Sample name | Date | Soil type | Sampling Sites | Latitude (°N) | Longitude (°W) | Nº soil samples | Nº aerosol samples |
|---|---|---|---|---|---|---|---|
| Nopal | 29/09/19 | Andosol | CDMX | 19.1991 | 99.0170 | 1 | 0 |
| Corn | 16/09/19 | Andosol | MOR | 19.0019 | 98.9161 | 1 | 0 |

| | | | | | | | |
|---|---|---|---|---|---|---|---|
| Bean | 24/02/20 | Calcisol | ZAC 1 | 22.8050 | 102.6750 | 1 | 1 |
| Chili | 25/02/20 | Calcisol | ZAC 2 | 22.8380 | 102.6853 | 1 | 1 |
| Wheat | 26/02/20 | Calcisol | ZAC 3 | 22.8508 | 102.6476 | 1 | 1 |
| Onion | 27/02/20 | Calcisol | ZAC 4 | 22.8164 | 102.6791 | 1 | 1 |
| Corn 2 | 14/10/20 | Leptosol | YUC | 20.9999 | 89.8575 | 1 | 0 |

The soil type was derived from SAGARPA (2017).

**Table S1:** Summary of the average meteorological conditions observed during the Zacatecas sampling campaign. The samples were collected between 4 h and 6 h.

| Sample name | Date | Latitude/ Longitude | Temperature (°C) | Relative humidity (%) | Radiation W/m$^2$ | Wind speed (km/h) |
|---|---|---|---|---|---|---|
| Bean | 24/02/2020 | 22.8050°N 102.6750°W | 21.79 | 21.48 | 778.13 | 18.50 |
| Chili | 25/02/2020 | 22.8380°N 102.6853°W | 16.52 | 36.80 | 483.32 | 12.64 |
| Wheat | 26/02/2020 | 22.8508°N 102.6476°W | 16.10 | 20.22 | 760.41 | 15.67 |
| Onion | 27/02/2020 | 22.8164°N 102.6791°W | 17.49 | 37.80 | 763.87 | 9.39 |

2) When did you collect the soil samples used for microbiological analysis and then when these were analyzed? If the authors would like to measure the colony forming units (CFUs) per gram, the analysis should be performed just after sampling, because the microorganisms in soils would be significantly changed once these were collected and stored at room temperatures. If the samples were not analyzed immediately, the data would not be scientifically valuable, and all the data and description about the CFU data (Lines 179-185; Lines 264-275; Figure S4) should be removed.

A: The Zacatecas soil samples were collected between 24/02/2020 and 27/02/2020 and the solutions for the analysis of microorganisms were immediately performed. Other soil samples were not treated immediately and is the reason why no biological analysis was performed on them.

3) I could not understand why only the INP data obtained from the MOUDI stages 3 to 6 (Figures 3 and 5) are reported, despite the facts that the samples were likely to be collected on the eight stages of the MOUDI (Lines 130-133). Although the authors explain that "the present results focus on particles >0.56 μm as it has been shown that particles >0.5 μm have a higher potential to act as INPs (e.g., DeMott et al., 2010) (Lines 190-191)", I think that they could evaluate whether the particles larger than 0.5 μm indeed have a higher potential to act as INPs than those smaller than 0.5 μm based on their data collected on the MOUDI stages 7 and 8. It is also unclear why the INP data larger than 5.6 μm (MOUDI stages 1 and 2) are excluded.

A: Typically, the number concentration of particles impacted on the glass coverslips placed in stages 1 and 2 is very low, so it inhibits the proper formation of droplets (a longer sampling time would be required to collect more particles in these stages). An opposite behavior is experienced on stage 8, as it usually gets overpopulated with particles that also inhibits droplet formation. The reviewer is correct in pointing out that we could have analyzed stage 7; however, based on previous studies using the MOUDI-DFT (e.g., Mason et al., 2016; Ladino

et al., 2019; Córdoba et al., 2021), we decided to focus on particles > 500 nm as supermicron particles were reported more likely to act as INPs.

4) In addition, I would like to suggest showing the total INP data (i.e., the sum of INP data obtained from all the MOUDI stages).

A: Thank you for this suggestion. The cumulative INP concentrations for the aerosol samples directly collected in the field were calculated as shown below in Figure A1. This information was added to the revised Figure 7 following the reviewer's suggestion.

[Figure]

**Figure A1:** Cumulative INPs concentration of aerosol samples collected in the field as a function of the freezing temperature.

5) A fatal flaw of this work is a lack of the reliability of the field INP data. As I already point out in Comment 1, it is not clear when and how the field samples were collected. However, if only 4 field INP data (only 1 field sample at each location) are available as shown in Figure 3, the numbers of the data are too small. Furthermore, it is also unclear whether the INP population in the field data were indeed well characterized by local agricultural soils, because their source and composition are not evaluated. Given the location of the sampling sites (Figure 1), there is the possibly that the aerosol population in the field INP data were characterized not only by local agricultural soils, but also by various types of aerosols from oceanic and urban/rural sources. If the authors would like to discuss the difference between the field and laboratory INP data based on only 4 field data, they should provide strong evidence that the available field data were mostly characterized by local agricultural soils. Otherwise, the discussion and conclusion about the comparison of the field and laboratory data presented in this manuscript (e.g., Lines 28-30; Lines 189-210; Lines 369-372) would not be scientifically valuable and hence should be removed.

A: The field aerosol samples were collected in a remote region only surrounded by agricultural fields. The largest town (Zacatecas City) in the area typically reports very low pollution levels (e.g., $PM_{2.5}$ ranging between 2.45 and 5.04 µg m$^{-3}$ during the sampling period). The closest ocean is 300 km away from the sampling site. Therefore, given these 3 facts mentioned, it is very unlikely that the field samples were heavily influenced by oceanic or urban emissions. As shown below in Figure A2, the aerosol concentration was much larger than the background concentration when the tractors were working close to the sampling sites. Although aerosol particles can be transported from long-distance sources into the sampling site, Figure A2 demonstrates that most of the aerosol particles collected on the glass cover slips during the field campaign were clearly generated by the local agricultural soils when they were being prepared.

[Figure]

**Figure A2:** Total particle concentration with the corresponding standard deviation measured during soil tillage at ZAC. The concentration of the aerosol particles was measured using a CPC 3010 (TSI) for dp > 30nm.

6) Although the authors describe that "the differences between laboratory and field environments are also reflected in different PSD observed during the aerosolization process (Fig. S1) (Lines 205-206)", I doubt if the particle size distribution (PSD) of the filed (F) samples would be totally characterized by local agricultural soils. Please show evidence that all the sizes (0.3-10 μm) of the field samples were entirely characterized by local agricultural soils.

A: During the field measurements the particle size distributions (PSD) were obtained with the LasAir for particle sizes ranging between 0.3-0.5 μm, 0.5-1.0 μm, 1.0-5.0 μm, 5.0-10 μm, and >10 μm. As shown below in Figure A3, PSD were obtained for two conditions (i.e., background and emission). Background measurements refer to precise times when tractors were not working on the soil, while measurements labeled Emission refer to samples collected when the soils were being tilled. As Figure A3 shows, an increase in the particle concentration can be observed for the different particle sizes (except for chili aerosol sample at particles sizes between 0.3-0.5 μm and wheat aerosol samples at particle sizes between 1.0-5.0 μm). Therefore, taking into account the increase observed in Figure A3 together the total particle concentration behavior shown in Figure A2, we are confident that the field samples correspond entirely to emissions from the local agricultural soils.

[Figure]

**Figure A3:** Particle size distributions (PSD) of the ambient particles measured during the Zacatecas field campaign. The PSD were obtained when the tractors were operating (particles resuspension) and when they were off (Background). The error bars show the standard deviation of the average particle concentration for each channel.

7) If the authors consider that "the highest particle concentration for the L samples was found for particles between 1.0 μm and 5.0 μm (Fig. S1a), while the F samples are enriched in smaller particles, i.e., 0.3 μm (Fig. S2b) (Lines 208-209)" and "the larger particles present in the L samples likely promoted ice nucleation at warmer temperatures (line 210)", the authors would need to evaluate the ice nucleating ability of the laboratory (L) samples in the size range smaller than 0.56 μm (see also Comment 3).

A: As mentioned in our answer to comment 3, typically the number concentration of ambient particles impacted on the glass coverslips placed in stage 8 is very large. This behavior was also observed on stage 7 in the laboratory samples. Therefore, the INP concentration of stages 7 and 8 could not be evaluated. We agree with the reviewer that text included in the original manuscript (lines 208-210) is not completely appropriate, and therefore, the text was modified as follows. Lines 214-218: "Furthermore, the highest particle concentration for the L samples was found for particles between 1.0 μm and 5.0 μm (Fig. S1a), while the F samples were enriched in smaller particles, i.e., 0.3 μm (Fig. S1b). The difference in PSD can affect the ice nucleating abilities of the two types of samples; however, more experiments are needed to evaluate the contribution of different particle sizes to the total INPs concentrations."

8) I cannot understand how Figures S2 and S3 are prepared. In addition, I cannot understand why the results of the agricultural dust particles in the laboratory (before heat treatment) are different between Figures S2 and S3. Please explain the details of these figures.

A: In order to prepare Figures S2 and S3 (S4 in the revised manuscript), the $T_{50}$ values of the Zacatecas aerosol and soil samples were calculated. Afterwards, the $T_{50}$ values were grouped as a function of their particle size (stages 3 to 6) to make the box plots. These figures provide information about the median $T_{50}$, and the bottom and top edges of the box indicate the 25th and 75th percentiles, respectively. The whiskers represent the extreme data points and the + the outlier's values. We agree with the reviewer that the results for laboratory samples before heating should be the same in Figure S2 and S3; however, during the DFT experiments a few samples got damaged due to technical problems as shown below in Tables A1 and A2. Therefore, the data sets used in Figures S2 and S3 (now S4) are slightly different, as only the samples with available data before and after heating were included in Figure S3 (now S4) for comparative purposes.

To clarify this, the following text was added into the revised Figure S4 caption. Line 53-54: "The corn sample data was excluded for S3, as well as the bean and chili samples data for S6."

Lines 28-30 in the supplementary material: "Figures S2 and S3 were made from the $T_{50}$ values from the different frozen fraction curves. The box plot figures contain information about the median $T_{50}$, the bottom and top edges of the box indicate the 25th and 75th percentiles. The whiskers represent the extreme data points and the red pluses the outlier values."

**Table A1:** Summary of the $T_{50}$ data observed for laboratory and field samples.

| Sample | Laboratory samples | | | | Field samples | | | |
|--------|------|------|------|------|------|------|------|------|
|        | S3   | S4   | S5   | S6   | S3   | S4   | S5   | S6   |
| Corn   | -18.9 | -18.9 | -18.4 | -20.1 | - | - | - | - |
| Nopal  | -20.1 | -19.7 | -19.6 | -20.1 | - | - | - | - |
| Corn 2 | -20.0 | -19.7 | -20.0 | -20.3 | - | - | - | - |
| Bean   | -21.4 | -19.7 | -20.4 | - | -33.7 | -21.8 | -24.9 | -29.6 |

| | | | | | | | | |
|---|---|---|---|---|---|---|---|---|
| Chili | -21.6 | -19.4 | -20.4 | -20.6 | -24.7 | -24.3 | -27.3 | -26.3 |
| Wheat | -23.2 | -21.8 | -22.3 | -22.8 | -22.6 | -24.1 | -25.3 | -25.8 |
| Onion | -20.7 | -18.7 | -22.4 | -20.6 | -21.9 | -22.8 | -22.5 | -24 |

**Table A2:** Summary of the $T_{50}$ data observed for laboratory aerosol samples before and after the heating treatment.

| | Non heating samples (laboratory) | | | | Heating samples | | | |
|---|---|---|---|---|---|---|---|---|
| Sample | S3 | S4 | S5 | S6 | S3 | S4 | S5 | S6 |
| Corn | - | -18.9 | -18.4 | -20.1 | - | -26.5 | -23.8 | -24.7 |
| Nopal | -20.1 | -19.7 | -19.6 | -20.1 | -26 | -24.9 | -23.3 | -23.0 |
| Corn 2 | -20 | -19.7 | -20.0 | -20.3 | -31.7 | -31.4 | -34.0 | -29.2 |
| Bean | -21.4 | -21.9 | -20.4 | - | -23.3 | -24.9 | -26.7 | - |
| Chili | -21.6 | -19.4 | -20.4 | - | -27.9 | -29.0 | -25.4 | - |
| Wheat | -23.2 | -21.8 | -22.3 | -22.8 | -23.5 | -25.0 | -24.6 | -25.1 |
| Onion | -20.7 | -18.7 | -22.4 | -20.6 | -21.4 | -22.7 | -24.6 | -21.9 |

9) Previous studies (Tobo et al., 2014; Steinke et al., 2016) have reported the INAS of agricultural soil dust, instead of $T_{50}$ values. If the authors think that "the ice nucleation temperatures observed in the present study are on the same order as those reported for agricultural dust in Wyoming (USA), from -18ºC to -36ºC for dp=0.6 µm (Tobo et al., 2014), and Argentina, China, and Germany from -11ºC to -26ºC for dp <5 µm (Steinke et al., 2016) (Lines 231-233)", the INAS of agricultural soil dust presented in the manuscript should be calculated.

A: Following the reviewer suggestion, a new figure (Fig. S3) was added to the supplementary information, and the following text was added to the main text of the revised manuscript. Lines 243-248: "As shown in Figure S3, the freezing temperatures and the surface size active density ($n_S$) observed in the present study for aerosol particles collected in the field are on the same order as those reported for agricultural dust in Wyoming (USA), from $5.2 \times 10^4$ (-17°C) to $3.5 \times 10^7$ (-35°C) (Tobo et al., 2014). However, the $n_s$ values observed here are lower (by more than two orders of magnitude) than those reported for Argentina, China, and Germany at temperatures ranging between -11ºC and -26ºC for dp <5 µm (Steinke et al., 2016)."

10) Although this work tries to discuss the possible influence of organic matter (Section 3.2) and mineralogy, especially K-feldspar (Section 3.3) using several approaches, it is hard to understand the key conclusion of this study. Do you think which component would have more impact on the ice nucleating ability of the agricultural soils? Please discuss this point and explain in the Abstract and Conclusion sections.

A: Thank you for pointing this out. Based on the decrease of the ice nucleating ability after the heating treatment, the organic components seem to have a predominant role in the efficiency as INPs of the aerosol particles analyzed herein. However, it must be considered that the mineral and organic components suggest a size dependency. To clarify this, the following text was added to the revised manuscript.

Lines 34-36: "Finally, a decrease in the INPs efficiency after heating the samples at 300°C for 2 h, indicates that the organic matter from agricultural soils plays a predominant role in the ice nucleating abilities of this type of aerosol samples."

Lines 404-405: "Therefore, the organic components seem to have a predominant role in the efficiency as INPs of the aerosol particles analyzed herein."

11) Although the correlation map is shown in Figure S5, it is still hard to imagine the relationship of $T_{50}$ values with OC and mineral compositions for each sample. I would like to suggest preparing some scatter plots that compare these parameters (e.g., INPs vs. OC, K-feldspar, etc.).

A: The INP concentrations of the laboratory samples were very high compared to those of the field samples and are not atmospherically relevant. Therefore, we consider that the $T_{50}$ can be used as a metric to evaluate the ice nucleating behavior of the different samples. The median freezing temperature is likely linked with the composition of the aerosol samples, thus, having $T_{50}$ vs. OC and $T_{50}$ vs. K-feldspar etc. is a valid approach. Following the reviewer suggestion, we made the scatter plots (Figure A4 below) of the mineral composition and OC content against $T_{50}$, as a function of aerosol particle size. We respectfully consider that Figure S5 (now S6) is more quantitative than Figure A4, therefore, we decided to keep Figure S5 (now S6) as in the original manuscript.

[Figure]

**Figure A4:** Scatter plots between $T_{50}$, the concentration of K-feldspar, Plagioclase, Quartz, Kaolinite, and the OC concentration for particle ranging between 3.2 and 5.6 µm (S3), 1.8 and 3.2 µm (S4), 1.0 and 1.8 µm (S5), and 0.56 and 1.0 µm (S6).

12) In addition, I would like to suggest comparing the total INP data (see also Comment 4) with OC and mineral compositions, because the OC and mineral contents reported here would be based on the analysis of the bulk samples smaller than 10 µm. This result might be helpful to answer Comment 10 (which component would have more impact on the ice nucleating ability of the agricultural soils?).

A: Unfortunately, the mineralogical and OC analyses were performed on the laboratory samples only. Therefore, there is no data available to show this information. However, based on the heating treatment results from the laboratory samples and the positive correlation

between the $T_{50}$ and the OC concentration, the role of the organic components is highlighted in the revised manuscript (See comment 10).

**Technical Corrections**

13) Line 22: ice crystals formation => ice crystal formation

A: Changed as recommended.

14) Line 62: improving => enhancing or influencing (or something like this)

A: "improving" was changed to "enhancing"

15) Lines 68-69: Garcia et al. (2012) reported the results from aerosol sampling in the air and not in soils.

A: "aerosol samples" was added instead of "soils" in the revised manuscript

16) Lines 95 and 104: The terminology like Morelos (M-MOR) and Morelos (M-ZAC) are confusing.

A: Unfortunately, there is a city in the Zacatecas State called Morelos, but there is also a State called Morelos. To avoid confusion, the terminology was modified in Table 1, and the sampling locations were renamed taking into account the States such as CDMX, MOR, ZAC 1, ZAC 2, ZAC 3, ZAC 4, and YUC. We hope this new terminology avoids the previous confusion.

17) Line 193: solid lines => solid curves?

A: Changed as recommended.

18) Line 194: dotted lines => dashed curves?

A: Changed as recommended.

19) Line 209: Fig. S2b => Fig. S1b?

A: Corrected.

20) Line 243: dp <0.6 μm => dp = 0.6 μm

A: Changed as recommended.

21) Line 301: What is "Hunucmá sample"?

A: Hunucma sample refers to the sample corn 2, which was collected at a location called Hunucma. The text was modified to avoid confusion as follows. Line 320: "feldspars compounds in the corn 2 sample".

22) Table 1: Please clearly explain why the samples are labeled as "Nopal", "Corn", "Bean", "Chili", "Wheat", "Onion", and "Corn 2" (please add some explanations in the Methods section). In addition, what do you mean by "°O" and "N°"?

A: The labels of the samples are now explained in the revised manuscript. Lines 100-101: "The location of the sampling sites and the number of samples collected are summarized in Table 1, where samples are labeled based on the previous crop present at each site."

We apologize for the °O and N°. They refer to West and North coordinates. "O" was replaced by "W".

23) Figure 1: It is hard to see the location of each sampling point, because the same symbols with similar colors are used.

A: The color of the symbols was modified in the revised manuscript to avoid confusion.

24) Figure 2: It is hard to understand the experimental setup. More detail schematic images (particularly, around aerosol collector) should be presented.

A: The diagram of the experimental setup was modified in the revised manuscript to avoid confusion. The aerosol collector represents the different sampling aerosol instruments that were used in order to collect the aerosol samples. To perform the analysis of INPs, a cascade impactor MOUDI was used, for the mineralogical and OC analysis a MiniVol was used, and to determine the PSD an OPC was used.

25) Figure 3: It is hard to see the difference of curves, because similar curves and colors are used. Please prepare more eye-friendly figures.

A: The colors of the curves in Figure 3 were modified in the revised manuscript based on the reviewer's suggestions.

26) Figures 3 and caption: d) 0.56.1.0 µm => 0.56-1.0 µm

A: The Figure captions were corrected in the revised manuscript.

27) Figure 4 caption: Organic carbon, elemental carbon, and mineral contribution ~.

A: The figure caption was modified in the revised manuscript following the reviewer's suggestion.

28) Figure 5: It is hard to see the difference of curves, because similar curves and colors are used. Please prepare more eye-friendly figures.

A: The colors of the curves in Figure 5 were modified in the revised manuscript based on the reviewer's suggestions.

29) Figure 7: Please explain how the red-colored range of "Field samples (this study)" is defined. In addition, the authors would need to prepare a supplementary figure showing the comparison of the INP data with this red-colored range. In this figure, the INP data should be the total INP number concentration (see also Comment 4).

A: The red area was defined by plotting the individual INPs concentrations for the four field aerosol samples for the 4 different size bins. Based on the reviewer's suggestion, Figure 7 was modified using the total INP number concentration and the following text was modified in the revised manuscript. Lines 349-350: "The total INP concentration emitted during soil tillage of the agricultural soils in ZAC was found to vary between 0.11 $L^{-1}$ and $10^2$ $L^{-1}$ from -15.01$^0$C to -34.5$^0$C as shown in Fig. 7."

30) Figure S1: I would suggest the use of dN/dlogD$_p$ instead of aerosol concentrations (N). In addition, please clearly explain when and how the data in Figure S1b were measured.

A: Figure S1 was modified as suggested. Additionally, the following text was added in the revised manuscript. Lines 138-141: "Additionally, an Optical Particle Counter (OPC) LasAir III

(310 B; Particle Measuring Systems) was used to obtain the particle size distribution (PSD) of both the aerosol samples generated in the laboratory and those measured in-situ during the field campaign. The OPC was operated at a flow rate of 28.3 L min$^{-1}$, and the aerosol concentrations correspond to particle sizes ranging between 0.3 μm and 10 μm."

Line 213: "mean particle concentrations between $1.3 \times 10^{-3}$ and 1.2 particles cm$^{-3}$ characterized L samples."

31) Figures S2 and S3: Please explain what the box-and-whisker plots indicate.

A: The following text was added in the supplementary material of the revised manuscript. Lines 28-30: "The box plot figures contain information about the median $T_{50}$, the bottom and top edges of the box indicate the 25$^{th}$ and 75$^{th}$ percentiles. The whiskers represent the extreme data points and the red pluses the outlier values."

**References**

Córdoba, F., Ramirez, C., Cabrera, D., Raga, G. B., Miranda, J., Alvarez, H., Rosas, D., Figueroa, B., Kim, J., Yakobi-Hancock, J. D., Amador, T., Gutierrez, W., García, M., Bertram, A., Baumgardner, and Ladino, L. A.: Ice nucleating abilities of biomass burning, African dust, and sea spray aerosol particles over the Yucatan Peninsula, Atmos. Chem. Phys., 21(6), 4453–4470, doi:10.5194/acp-21-4453-2021, 2021

Ladino, L. A., Raga, G. B., Alvarez-Ospina, H., Andino-Enríquez, M. A., Rosas, I., Martínez, L., Salinas, E., Miranda, J., Ramírez-Díaz, Z., Figueroa, B., Chou, C., Bertram, A. K., Quintana, E. T., Maldonado, L. A., García-Reynoso, A., Si, M., and Irish, V. E.: Ice-nucleating particles in a coastal tropical site, Atmos. Chem. Phys., 19, 6147–6165, doi:10.5194/acp-19-6147-2019, 2019.

Mason, R. H., Si, M., Chou, C., Irish, V. E., Dickie, R., Elizondo, P., Wong, R., Brintnell, M., Elsasser, M., Lassar, W. M., Pierce, K. M., Leaitch, W. R., MacDonald, A. M., Platt, A., Toom-Sauntry, D., Sarda-Estève, R., Schiller, C. L., Suski, K. J., Hill, T. C. J., Abbatt, J. P. D., Huffman, J. A., DeMott, P. J., and Bertram, A. K.: Size-resolved measurements of ice-nucleating particles at six locations in North America and one in Europe, Atmos. Chem. Phys., 16, 1637–1651, doi:10.5194/acp-16-1637-2016, 2016.

**REFEREE 2**

**General comments:**

One conclusion reached by the authors is that agricultural soil dust contributes to the formation of mixed-phase clouds (line 368). However, the data presented in this manuscript do not fully support this conclusion. Specifically, transport measurements were not part of this study and consequently, the influence of the sampled aerosol on cloud formation can only be assumed. Indeed, it could be that these soil dusts are transported to altitudes where mixed-phase clouds appear and the INP concentrations from this study match with literature data, however more data is needed to proof this hypothesis in this context. Therefore, I propose to weaken the according argument in the conclusion. The manuscript could be improved if the authors deepened the discussion of possible transport processes of soil from fields to high altitudes.

A: Thank you for highlighting this point. The transport processes of aerosol particles derived from agricultural soils to high altitudes was not the focus of this study. Therefore, the conclusion was modified as follows. Line 386-388: "The measured concentrations are comparable to those reported for agricultural soils in the United States and England, confirming the contribution from tropical emissions and the potential they have to impact mixed phase cloud formation if brought to altitudes higher than cloud base."

The authors use freezing curves to depict the INA of their aerosol samples. However, in the discussion the authors focus the interpretation rather on $T_{50}$ values than the freezing curves themselves. $T_{50}$ values have been used in the scientific community to represent a sample's INA. For pure ice nuclei that freeze immediately at a given temperature, $T_{50}$ is a good parameter for comparison studies. However, if one sample consist of more than one ice nucleus (e.g. OC and K-feldspar), freezing curves may show steps in the spectrum (see e.g. onion (F), wheat (F) in Figure 3). Consequently, the $T_{50}$ values does not fully represent such samples. Have the authors considered also taking $T_{10}$ and $T_{90}$ into consideration or include a more detailed discussion about the spectra?

A: Thank you for this suggestion. As shown below in Figure A5, the $T_{10}$ and $T_{90}$ behavior for the laboratory and field samples is similar to the $T_{50}$ behavior. On average, the laboratory samples have higher freezing temperatures (> 2°C) than those observed in field samples. Although the $T_{10}$ and $T_{90}$ values corroborate the $T_{50}$ results, we decided not to include/discuss this information in the revised manuscript as usually $T_{50}$ is considered as a statistically significant metric to define the ice nucleation abilities of aerosol particles (Kanji et al., 2017).

[Figure]

**Figure A5:** Average $T_{10}$ (left) and $T_{90}$ (right) of the aerosol samples collected at the field (F, red boxes) and those generated in the laboratory (L, grey boxes) for particles ranging between 3.2 and 5.6 µm

(S3), 1.8 and 3.2 μm (S4), 1.0 and 1.8 μm (S5), and 0.56 and 1.0 μm (S6). The red cross indicates an outlier value of the $T_{10}$.

**Specific comments:**

**Abstract:**

Line 32: The authors state that $T_{50}$ values and aerosol particle size are correlated. There is no mention of what exactly is meant by particle size. I assume aerodynamic diameter? I would recommend mentioning this in the text.

A: The text was modified in the revised manuscript as follows. Line 32: "particles with aerodynamic diameters between 1.8 μm and 3.2 μm"

Line 33: Please indicate precisely which efficiency is meant. Ice nucleation efficiency?

A: The text was corrected following the reviewer's suggestion. Line 33: "ice nucleation efficiency of aerosol samples"

**Introduction:**

Line 61: not all bacteria and fungi are ice nucleation active. I would recommend to add 'certain' to the sentence: '[…] (e.g. certain bacteria, fungi) […]'

A: The word "certain" was added to the sentence in the revised manuscript.

**Methods:**

Line 111: I was wondering what the weather conditions were like during the campaign? Did the authors record any data on the meteorological conditions?

A: Thank you for pointing this out. The average meteorological conditions observed during the Zacatecas field campaign are summarized in the following table, which was added to the revised Supplementary Material as Table S1.

Table S1. Summary of the average meteorological conditions observed during the Zacatecas sampling campaign. The samples were collected between 4 h and 6 h.

| Sample name | Date | Latitude/ Longitude | Temperature (°) | Relative humidity (%) | Radiation W/m$^2$ | Wind speed (km/h) |
|---|---|---|---|---|---|---|
| Bean | 24/02/2020 | 22.8050°N 102.6750°W | 21.79 | 21.48 | 778.13 | 18.50 |
| Chili | 25/02/2020 | 22.8380°N 102.6853°W | 16.52 | 36.80 | 483.32 | 12.64 |
| Wheat | 26/02/2020 | 22.8508°N 102.6476°W | 16.10 | 20.22 | 760.41 | 15.67 |
| Onion | 27/02/2020 | 22.8164°N 102.6791°W | 17.49 | 37.80 | 763.87 | 9.39 |

Line 111: '[…] samples collected at the ground level.' How much distance was between the instruments and the ground? Did they actually stand on the ground or were the instrument mounted onto something?

A: The samples were collected at ground level. However, they were not directly placed over the ground. The inlet of the instruments was at approximately 1.5 m a.g.l. This information was added to the revised manuscript in Lines 113-114.

Line 116: Is there a specific reason why aerosol and soil samples were stored at different temperatures?

A: The soil samples were stored at room temperature following the Mexican regulations commonly applied for analysis of soils (NMX-AA-132-SCFI-2016°), while the aerosol samples were stored at 3°C as a common storage protocol (e.g., Mason et al., 2016; Ladino et al., 2019; Córdoba et al., 2021).

Figure 2: I was wondering if the 'OPC sign' shouldn't be indicated before the 'aerosol collector sign'? Or was the OPC attached after the impactor/MiniVol?

A: We apologize for the confusion. The OPC was not attached after the aerosol collector. The diagram indicates that the same sampling system was used to collect the aerosol samples and also to measure the aerosol concentration. The figure was modified in the revised manuscript.

Line 115 and line 123: For how long did you collect your aerosols? Was the time period in the field the same as in the laboratory?

A: The sampling time for the ambient particles in the field varies between 4 and 6 h (now added to Table S1), while the samples generated in the laboratory were collected between 20 to 80 seconds as they were highly concentrated.

Line 130: Did the authors collect the aerosol samples for the ice nucleation and mineral analysis in parallel or were the measurements performed sequentially?

A: The aerosol samples for ice nucleation and mineralogical analysis were collected sequentially as different filter types were used. For the INP analysis glass coverslips were used, and once the INP samples were collected, aluminum filters were placed in the MOUDI stages to collect a new sample for the mineralogical analysis.

Line 146: Are the authors referring to the diameter of the droplets (d=170 um) as the size?

A: Yes, the manuscript refers to droplets' diameter. The text was corrected as follows. Line 151: "until the droplets have reached a diameter of 170 μm (on average)".

Line 147: I would recommend writing 'dry nitrogen', rather than dry air.

A: "dry nitrogen" was added to the revised manuscript.

Line 147: I assume that the droplets were not evaporated completely but rather reduced to the desired size? Please clarify.

A: The reviewer is correct; the droplets were not completely evaporated. Once the droplets reach a diameter around 170 μm, dry nitrogen is used to shrink the droplets to avoid contact between them. Further details of the experimental procedure are mentioned in Cordoba et al. (2021).

The text was corrected as follows: Lines 150-153: "Afterwards, humid air was directed towards the sample to allow liquid droplet formation over the aerosol particles, until the droplets have

reached a diameter of 170 µm (on average). Once most of the droplets have reached this size, dry nitrogen was used to shrink the size of the droplets to avoid contact between them."

Line 152: Could the authors give a little bit of insight into the statistics of their setup? How many droplets/particles per glass slide were analyzed for one sample?

A: The number of droplets/particles vary depending on the sample. However, on each glass slide approximately 20 to 30 droplets are usually formed. Then, the concentration of ice nucleating particles was computed using Mason et al. (2015) equations as explained in the manuscript. The following text was added to the revised manuscript. Line 158: "The number of droplets formed during each experiment on an individual glass slide varied between 20 and 30."

Line 181: This sentence is incomplete. What is meant by 'at 0.85%'? Does that refer to a concentration of the buffer? Further, could the authors specify which sterile solution they used? In addition, was the cultivation performed at room temperature? Maybe the authors could mention that in the text.

A: For the biological cultures a sterile physiological saline solution at 0.85% (w/v of NaCl) was used. In the case of bacteria, they were incubated at 35°C during 48 h, and fungi at 25°C for 72 h. The text was corrected as following in the main manuscript, lines 188-191: To determine culturable microorganisms present on the soil samples collected in ZAC, 500 mg of each sample were added in 10 mL of sterile solution at 0.85% (w/v of NaCl). After 1:100 dilution and vortex agitation, 0.1 mL of solutions were cultured on three growing media such as Trypticase Soy Agar (TSA) and MacConkey Agar (MCA) between 24 and 48 h at 35°C, and Malt Extract Agar (MEA) for 3 days at 25°C.

**Results and Discussion:**

Figure 3: For clarification to the reader, I would suggest to mention the aerodynamic diameter as the size in the caption.

A: The Figure caption was modified as suggested by the reviewer.

Figure S1: It is not clear to me how the particle size distribution was recorded for the field measurements. Could the authors please state in the 'Method' section how the measurement was performed?

A: The particle size distributions (PSD) for the aerosol samples collected in the field were also derived from the optical particle counter LasAir III. The text was modified in the revised manuscript as follows. Lines 138-141: "Additionally, an Optical Particle Counter (OPC) LasAir III (310 B; Particle Measuring Systems) was used to obtain the PSD of both the aerosol samples generated in the laboratory and those measured in-situ during the field campaign. The OPC was operated at a flow rate of 28.3 L min$^{-1}$, and the aerosol concentrations correspond for particle sizes ranging between 0.3 µm and 10 µm."

Line 212: I was impressed that the type of crop which previously grew on the field influenced the ice nucleation ability of the soil. Do you think field treatments (e.g., pesticides, fertilizer or no artificial treatments that could promote biodiversity and possibly ice-active microbes) could affect the INA of the soil? In addition, do you think that freeze tolerant plants (see e.g. Marcellos and Single, 1979) may leach ice nuclei into the soil?

A: As the presence of fertilizers or additives on soils have been proposed to influence the organic content (ex. humic substances) or soil properties (Martin et al., 1966; Peña-Méndez et al., 2005; Suski et al., 2018), they would likely influence the emission of aerosol particles

from soils. Therefore, their ice nucleating abilities can be influenced; however, more information is required to understand the nature of the soil components responsible for their ice nucleating activity. Following the observations of Marcellos and Single (1979), plant fragments may have supercooling abilities. However, it has been observed that freeze tolerant plants are able to influence ice crystal growth and those containing ice binding proteins can inhibit the ice nucleating activity of certain pathogens (Bredow & Walker, 2017). Therefore, it has been suggested that certain freeze tolerant plant species can suppress the ice nucleating abilities of soil particles. While we recognize the importance of understanding the role of plant residues in the INA of soil, such research is beyond the focus of the present study.

We apologize for the confusion. The types of crops previously present in the different sampling areas were suggested to influence the ice nucleating abilities of the soils as they are a source of organic matter. However, this analysis was beyond the scope of our study and not enough information is available to confirm this hypothesis. Therefore, the text in the revised manuscript was corrected as follows. Lines 220-226: "It was also found that the ice nucleating abilities of the different aerosol samples varies, with particles from the nopal, corn 1, and corn 2 samples showing the warmest freezing temperatures and with the beans and wheat samples showing the coldest freezing temperatures (Fig. 3). The presence of additives on soils have been proposed to influence the organic content or soil properties (Martin et al., 1966; Peña-Méndez et al., 2005; Suski et al., 2018), therefore, they could influence the emission of aerosol particles from soils. However, more information is required to understand the nature of the soil components responsible for their ice nucleating activity."

Figure 5: I would recommend to add the temperature and duration of the heat treatment to the caption.

A: Following the reviewer's suggestion, the Figure caption was modified as follows. "For the heating treatment, the samples were heated at 300⁰C for 2 h."

Figure 3 and 5: When printing out the manuscript the lines of Corn 2 (L), Bean (L) and Onion (L) are hard to distinguish. The authors may consider changing the colors.

A: The colors of the curves were modified in revised Figures 3 and 5.

Line 281: I recomend to also cite Zolles et al. (2015) here, as they showed that the INA of e.g. microcline decreased by 2 degrees after the sample was heated to 250°C.

A: Thank you for the suggestion. Zolles et al. (2015) was added to the revised manuscript with the following text. Lines 298-301: "Although those studies suggested that minerals are not strongly affected by dry heat treatments, Zolles et al. (2015) and Daily et al. (2021) showed that shifts in minerals efficiency as INPs cannot be neglected, as heat treatments at 250⁰C might influence the ice nucleating abilities of feldspar compounds."

**Technical remarks:**

Line 45: INP should be in plural (INPs)

Line 145: Missing the letter s: 'Afterwards, […]'

Line 640: The micrometer symbol is in a different style

A: The technical remarks were added into the revised manuscript.

**References**

Bredow, M., and Walker, V. K.: Ice-binding proteins in plants. Frontiers in plant science, 8, 2153, doi:10.3389/fpls.2017.02153, 2017.

Córdoba, F., Ramirez, C., Cabrera, D., Raga, G. B., Miranda, J., Alvarez, H., Rosas, D., Figueroa, B., Kim, J., Yakobi-Hancock, J. D., Amador, T., Gutierrez, W., García, M., Bertram, A., Baumgardner, and Ladino, L. A.: Ice nucleating abilities of biomass burning, African dust, and sea spray aerosol particles over the Yucatan Peninsula, Atmos. Chem. Phys., 21(6), 4453–4470, doi:10.5194/acp-21-4453-2021, 2021;

Kanji, Z. A., Ladino, L. A., Wex, H., Boose, Y., Burkert-Kohn, M., Cziczo, D. J., and Krämer, M.: Overview of ice nucleating particles, Meteor. Mon., 58, 1-1, doi:10.1175/AMSMONOGRAPHS-D-16-0006.1., 2017.

Martin, J. P.: Influence of pesticides on soil microbes and soil properties. Pesticides and their effects on soils and water, 8, 95-108, 1996.

Marcellos, H., Single, W.V.: Supercooling and heterogeneous nucleation of freezing in tissues of tender plants. Cryobiology, 16(1), 74–77, doi:10.1016/0011-2240(79)90013-0, 1979.

Peña-Méndez, E. M., Havel, J., & Patočka, J.: Humic substances-compounds of still unknown structure: applications in agriculture, industry, environment, and biomedicine. *J. Appl. Biomed*, *3*(1), 13-24, 2005.

Suski, K. J., Hill, T. C. J., Levin, E. J. T., Miller, A., DeMott, P. J., and Kreidenweis, S. M.: Agricultural harvesting emissions of ice-nucleating particles, Atmos. Chem. Phys., 18, 13755–13771, doi:10.5194/acp-18-13755-2018, 2018.